# Inequalities in healthcare use during the COVID-19 pandemic

Arun Frey [1,2,3,4,5,7], Andrea M. Tilstra [1,2,5,6,7] & Mark D. Verhagen [1,2,3,5,6,7]

The COVID-19 pandemic led to reductions in non-COVID related healthcare use, but little is known whether this burden is shared equally. This study investigates whether reductions in administered care disproportionately affected certain sociodemographic strata, in particular marginalised groups. Using detailed medical claims data from the Dutch universal health care system and rich full population registry data, we predict expected healthcare use based on pre-pandemic trends (2017 – Feb 2020) and compare these expectations with observed healthcare use in 2020 and 2021. Our findings reveal a 10% decline in the number of weekly treated patients in 2020 and a 3% decline in 2021 relative to prior years. These declines are unequally distributed and are more pronounced for individuals below the poverty line, females, older people, and individuals with a migrant background, particularly during the initial wave of COVID-19 hospitalisations and for middle and low urgency procedures. While reductions in non-COVID related healthcare decreased following the initial shock of the pandemic, inequalities persist throughout 2020 and 2021. Our results demonstrate that the pandemic has not only had an unequal toll in terms of the direct health burden of the pandemic, but has also had a differential impact on the use of non-COVID healthcare.

The COVID-19 pandemic (henceforth: pandemic) has affected healthcare services across the world, placing systems under considerable strain, contributing to delayed or even missed care and treatment[1-7]. The overall declines in healthcare use are well recorded[8], as is knowledge of the unequal burden of the direct effects of the pandemic on marginalised populations[9-16]. Less is known of how the collateral effects of the pandemic have affected different sociodemographic groups. We address this gap by analysing detailed information on healthcare expenditures and rich registry data on all individuals registered in The Netherlands between 2017 and 2021, quantifying the number of missed healthcare treatments for different population groups.

Beyond elevated rates of mortality from COVID[17,18], and the emerging consequences of long COVID[19-21], the pandemic has also disrupted non-COVID healthcare services. Healthcare systems across the world responded to the imminent threat of the pandemic by changing their institutional practices in order to anticipate increased hospitalisations due to COVID, necessitating a decline in the availability of non-COVID healthcare procedures[3,5,6]. This decline was especially pronounced for procedures deemed nonessential or elective, as hospitals and public health authorities made triaging decisions over which treatments to prioritise[1,2,4,7]. It is unclear whether the reduction in care was evenly distributed among all individuals or if specific population groups were disproportionately affected. Insights into how the decline of non-COVID healthcare differed between sociodemographic groups remain an important line of inquiry that can inform policy efforts to mitigate the long-term effects of missed or delayed healthcare[4,22-25].

[1]Leverhulme Centre for Demographic Science, 42 Park End St, Oxford OX1 1JD, UK. [2]Nuffield College, University of Oxford, 1 New Rd, Oxford OX1 1NF, UK. [3]Amsterdam Health and Technology Institute, Paasheuvelweg 25, Amsterdam 1105 BP, The Netherlands. [4]Stanford University, 450 Jane Stanford Way, Stanford, CA 94305, USA. [5]Department of Sociology, University of Oxford, 42 Park End St, Oxford OX3 7LF, UK. [6]Nuffield Department of Population Health, University of Oxford, 42 Park End St, Oxford OX1 1JD, UK. [7]These authors contributed equally: Arun Frey, Andrea M. Tilstra, Mark D. Verhagen. ✉e-mail: mark.verhagen@nuffield.ox.ac.uk

There are various reasons to expect group differences in the decline of non-COVID healthcare procedures. First, the pandemic likely changed the underlying need for certain medical procedures. The pandemic and subsequent lockdown led to drastic behavioural changes that altered pre-existing patterns of illnesses and injuries. Traumatic injuries following traffic accidents, for example, declined considerably during lockdown, as individuals spent less time commuting or travelling[26]. Such injuries are more common for some population groups than others, so changes to health needs may have contributed to differences in healthcare use between population groups. The need for other procedures, such as cancer treatments, should have remained largely constant during the pandemic.

Second, people may have avoided visiting healthcare facilities during the pandemic, resulting in missed care[8,27–29]. In The Netherlands, there are notable patterns in care avoidance during the pandemic: older individuals, females, unemployed people, and people in poorer health reported the highest rates of healthcare avoidance[28]. Since the risk of adverse health effects following COVID infection differs by sociodemographic group, such fears may have disproportionately dissuaded some population groups from seeking the non-COVID care they needed, contributing to differences in healthcare use.

Third, sociodemographic groups may differ in their ability to successfully navigate a healthcare system under pressure. It is already known that marginalised people frequently experience barriers to accessing the same quality healthcare as their more advantaged peers, a pattern that can manifest through structural barriers (e.g., distance to healthcare facilities), financial barriers (e.g., cost of healthcare), or health beliefs and literacy[30,31]. This is especially the case with private healthcare systems, such as the United States, but inequalities in access have also been documented in public healthcare systems that, in theory, should provide equal access to all individuals[32–34]. The strong reductions in the availability of non-COVID healthcare may have further exacerbated such inequalities, as procedures became more scarce and harder to access.

In this paper, we quantify the weekly decline in non-COVID healthcare use during the first two years of the pandemic (2020 and 2021) and compare declines across several sociodemographic characteristics: age, sex, migrant background, and poverty status. We also stratify analyses by urgency level. In doing so, we examine whether the decline in healthcare use disproportionately affected certain subpopulations. To do this, we rely on rich population-level data from The Netherlands that we match to the universe of individual healthcare activities in the period 2017–2021 using full population insurance data from the Dutch universal health insurance system. We predict the expected number of individuals using healthcare based on pre-pandemic trends (2017 to February 2020) and compare these expectations with the observed number of individuals using healthcare

between March 2020 and December 2021. In doing so, we are able to assess both the magnitude and distribution of the decline in healthcare use during the first year of the pandemic for all individuals in The Netherlands.

The Netherlands presents a unique context for analysing inequalities in healthcare use during the pandemic, as healthcare coverage is universal and publicly available to all residents. The Netherlands is also characterised by i) small land mass, high population density, and dense infrastructure, ii) an efficient healthcare system and healthy population base, and iii) a wealthy population with comparatively low income inequality. This makes The Netherlands a case study where there is little reason to expect substantial inequalities in access to healthcare services. In line with most countries globally, the Dutch government also made concerted efforts to catch-up on postponed healthcare during the summer months of 2020−when the number of COVID cases was relatively mild−but subsequent rises in infections shortly thereafter have led to a considerable backlog[35]. As in many countries, catching up on missed care is still a central policy issue at the time of writing.

## Results

To estimate declines in healthcare use during the COVID-19 pandemic, we calculate the weekly number of unique patients treated in the Dutch healthcare system for the period between 2017 and 2021. We model this weekly count based on data from January 2017 through February 2020 and generate weekly predictions for the remainder of 2020 and 2021. Our analytic approach is summarised in Fig. 1, which shows the observed number of weekly treated individuals (black line) contrasted with the predicted number of treatments in the pre-pandemic (blue) and post-pandemic (red) periods. The close mapping of the blue and black lines in the pre-pandemic period demonstrates that our model fits the pre-pandemic trends well. We then compare the predicted weekly number of treated individuals when the pandemic started to the actually observed number of treated individuals (see Methods). We then repeat this analytical approach for various subsets of the population, as well as for medical subsets.

### Overall declines in non-COVID healthcare use

We estimate that on a weekly basis, about three million fewer patients received care by the end of the first year of the pandemic than expected. This decline in healthcare use persisted into the second year of the pandemic, with about one million fewer patients receiving care. Following the onset of the COVID-19 pandemic in March 2020, the number of weekly treated individuals declined rapidly (Fig. 2). At the height of the reduction at the end of March, 2020, more than 300,000 fewer individuals received care that week than expected, a near 45% decline in healthcare use. Although the number of individuals missing treatment decreased from mid-May onward, there was little to no catching up during the summer months, despite comparatively fewer COVID infections during this time and concerted policy efforts to catch up on missed care[35].

The observed decline in individuals receiving care following the onset of the pandemic is many times larger than the increase in treatments related to COVID infections. At the height of hospital use from COVID (the week of March 23rd, 2020, see Fig. 2), there were more than 6,300 additional individuals that received treatment for COVID infections, compared to a simultaneous decline in non-COVID related healthcare of more than 300,000 individuals. We observe a much less pronounced decline during the second peak of COVID hospitalisations at the end of 2020 and beginning of 2021, and again in the third peak of COVID hospitalisations towards the end of 2021, further underscoring the unique challenges of the early weeks of the pandemic and the considerable adaptation that occurred within the system.

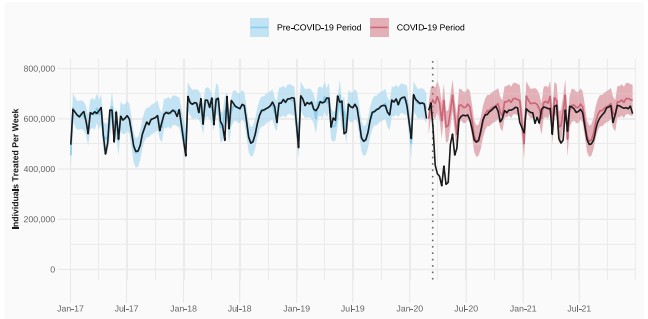

**Fig. 1 | Predicted and observed number of weekly patients, 2017 to 2021.** Model predictions of the number of treated individuals per week are depicted in blue (training) and red (predicted) lines (with 95% confidence intervals), observed realizations are depicted in black. Source data are provided as a Source Data file.

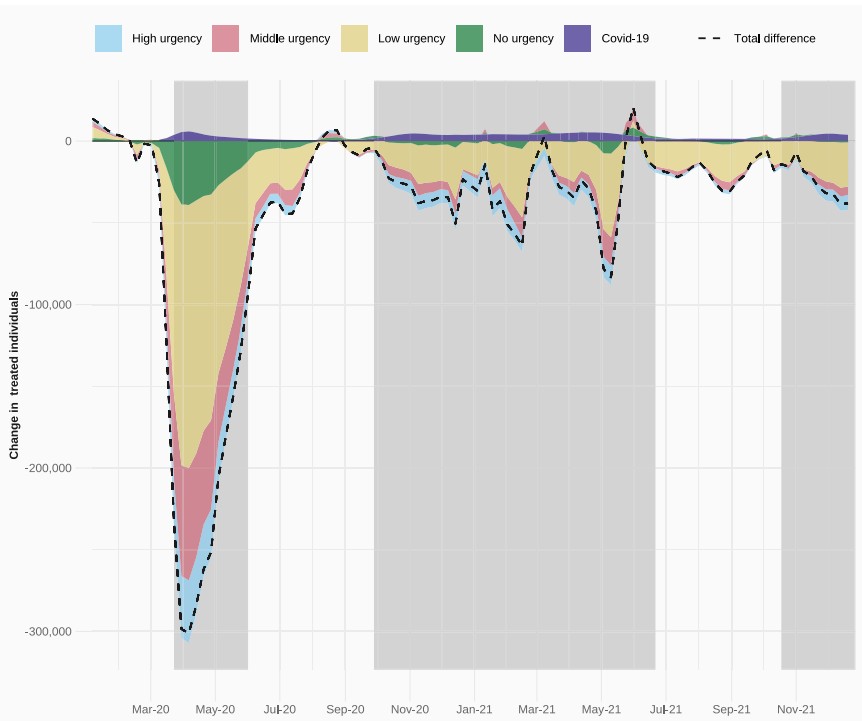

**Fig. 2 | Difference between the observed and predicted number of treated individuals per week in 2020 and 2021.** Colours differentiate between urgency types (high, middle, low, and unknown urgency) and patients treated for COVID-19. COVID hospital waves are depicted in shaded grey. Source data are provided as a Source Data file.

Most of the decline in healthcare procedures stems from those with low levels of urgency – i.e., procedures that do not have to be performed within a week. However, we also observe a considerable decrease in highly urgent activities that require more immediate medical attention (i.e., within a week): between March 2020 and December 2021, there were 5.2% fewer patients receiving highly urgent care, compared to 8.7% fewer patients receiving low urgency treatments.

## Sociodemographic inequalities in non-COVID healthcare use declines

Importantly, the observed decline in non-COVID healthcare use during the pandemic was not consistent across sociodemographic groups. In Fig. 3, we show the cumulative impact of the pandemic on healthcare use across different sociodemographic groups, which allows us to assess the magnitude of the uneven toll of the pandemic. To account for compositional differences, we age-sex standardise each sociodemographic group (see Methods). By the end of 2021, individuals living below the poverty line had experienced reductions in low and medium urgency procedures of 10.4% and 6.1%, respectively, compared to 9.0% and 4.3% for those above the poverty line. We document a similar trend for individuals with a migrant background, who experienced reductions of 10.0% and 4.7% in low and medium urgency procedures compared to 9.0% and 4.3% for individuals without a migrant background. Older people and females also saw greater declines in healthcare use than younger people and males, although inequality between sex reduced as the pandemic progressed. For highly urgent procedures, we find considerably lower levels of inequality, but continue observing a substantial age gradient in urgent health use: individuals aged 76 experienced declines of 7.6%, whereas younger patients (18 to 29) recorded declines of 4.3%. These changes in year-over-year health use are substantial, especially when considering that the COVID-19 pandemic did not impact healthcare use in January and most of February 2020.

These findings are further supported by multivariate regressions where we assess the joint impact of sociodemographic characteristics on weekly declines in non-COVID healthcare, as well as during different stages of the pandemic (see Figure SI-1). We find that associations between those below the poverty line and people with a migrant background operate independent from one another. Further interactions between our sociodemographic variables and pandemic time periods show that females had the strongest decreases during the initial phase of the pandemic. Whereas declines were considerable for all sociodemographic groups relative to pre-pandemic levels, females endured a further 20 percentage point decline in weekly treated patients relative to their pre-pandemic averages compared to males. From the second wave onwards, this pattern is reversed. Overall, the strongest inequality across the entire period up until December 2021 was experienced by people with a migrant background.

## Declines for trauma-related and oncological care

We explore whether the observed group differences are driven by underlying changes in the need for healthcare (see Methods). For example, much of the observed decline in highly urgent care during the first pandemic wave can be explained as a consequence of large reductions in trauma-related cases (Figure SI-2a and SI-2b). These reductions can partly be explained by behavioural changes stemming from government lockdowns. However, the pandemic also decreased highly urgent oncological care, with 5% fewer patients receiving critical oncological care by the end of 2020 than in previous years, even though the need for oncological procedures should not have changed as a result of the pandemic.

We show inequalities in the use of oncological (left) and trauma (right) healthcare (Fig. 4). This comparison allows us to disentangle the effect of behavioural interventions such as lockdowns on the use of care, and how these different mechanisms factor into our observed inequalities. In line with our expectations, we observe stronger overall reductions in trauma-related care than oncological care. In addition,

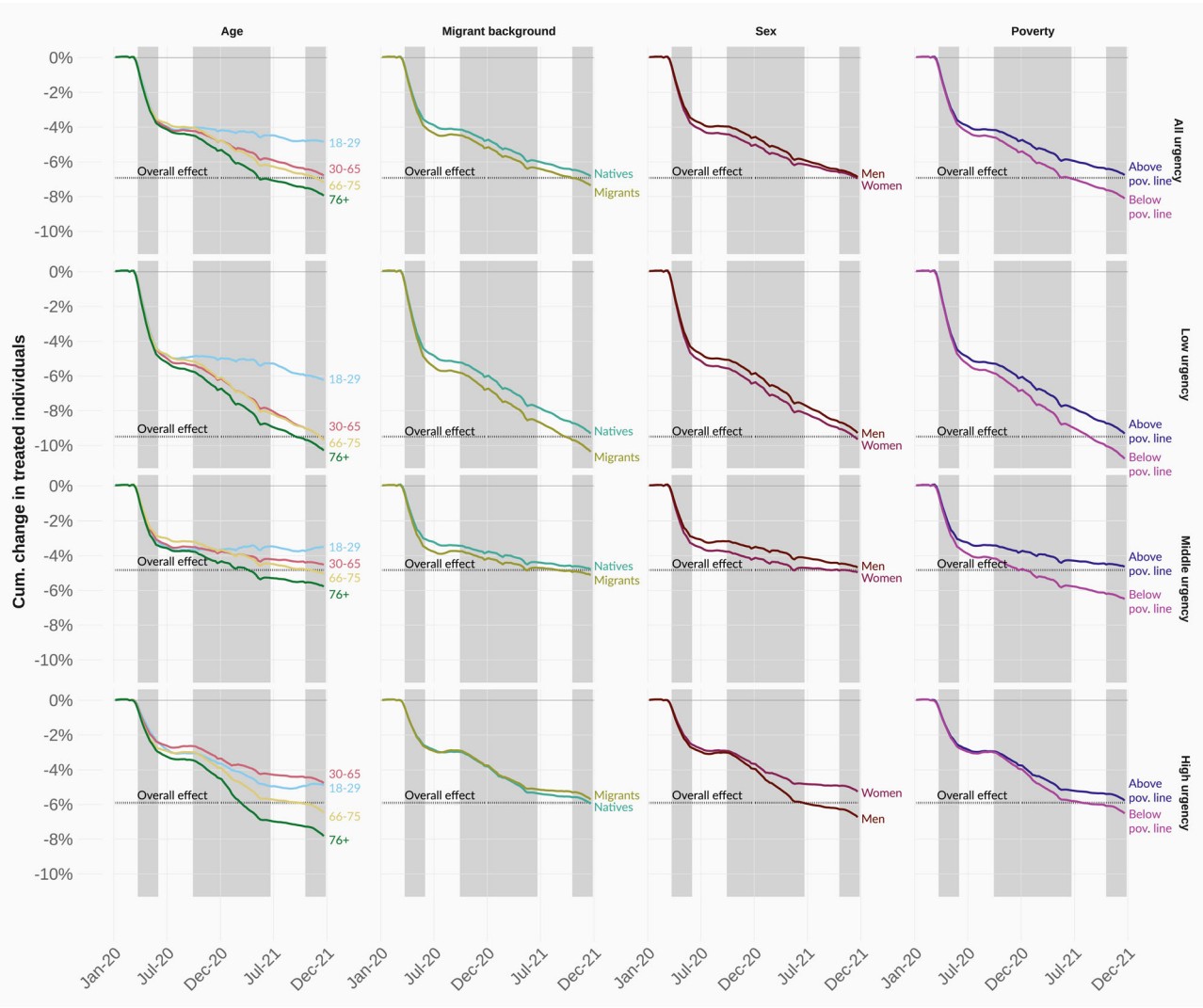

**Fig. 3 | Cumulative age- and sex-adjusted differences between the observed and predicted number of treated individuals in 2020 and 2021, across urgency** types (rows) and sociodemographic groups (columns). COVID hospital waves are depicted in shaded grey. Source data are provided as a Source Data file.

the strongest reduction in trauma-related procedures can be observed among young males, who were otherwise less impacted by reductions in healthcare procedures. However, we also document stark differences in the use of oncological procedures between sociodemographic groups, particularly for critical procedures. By the end of 2020, females were almost twice as likely to miss a critical oncological procedure compared to males. Older people (76 and above) recorded losses that were nearly twice the overall reduction. In fact, we observe a decline in oncological procedures across all population groups with the exception of highly urgent oncological procedures among 18-29-year-olds, for whom we even observe a slight increase by the end of 2021.

**Robustness checks**

The findings documented here are robust to a series of empirical robustness checks. These include subsetting the universe of healthcare procedures to i) exclude all diagnostic laboratory tests (Figures SI-4a and SI-4b), and to ii) only consider procedures that included a clinical or emergency room activity (Figures SI-5a and SI-5b). We also re-estimate results using the weekly number of performed healthcare activities as the unit of analysis, rather than the weekly number of individuals receiving care (Figures SI-6a and SI-6b). Across these

specifications, we find similar patterns in healthcare declines and inequalities during the pandemic.

The accuracy of our estimates rest on the extent to which predictions of healthcare use in 2020 would have mirrored observed healthcare use, had the pandemic not occurred. To gauge the validity of this assumption, we perform a placebo analysis where we set 2019 as the year of interest, and predictions closely mirror observed healthcare use that year (Figures SI-7a and SI-7b). We also assess whether our results are robust to using a Negative Binomial regression to make estimates of expected healthcare use instead of a standard linear regression and find similar results (SI-8a and SI-8b).

Finally, we assess the possibility that our observed findings are due to selective mortality from COVID. Given that the frail and older people were disproportionately likely to use the healthcare system and to die from COVID, mortality due to COVID may have reduced healthcare use that would have otherwise occurred. In Table SI-1 we display the number of COVID-19-related deaths by sociodemographic characteristics. In line with findings elsewhere, we observe a disproportionate number of deaths among older people, those living below the poverty line, and those with a migrant background. However, the number of deaths attributed to COVID accounts for less than 0.2% of the unique recipients of healthcare across 2020 and 2021,

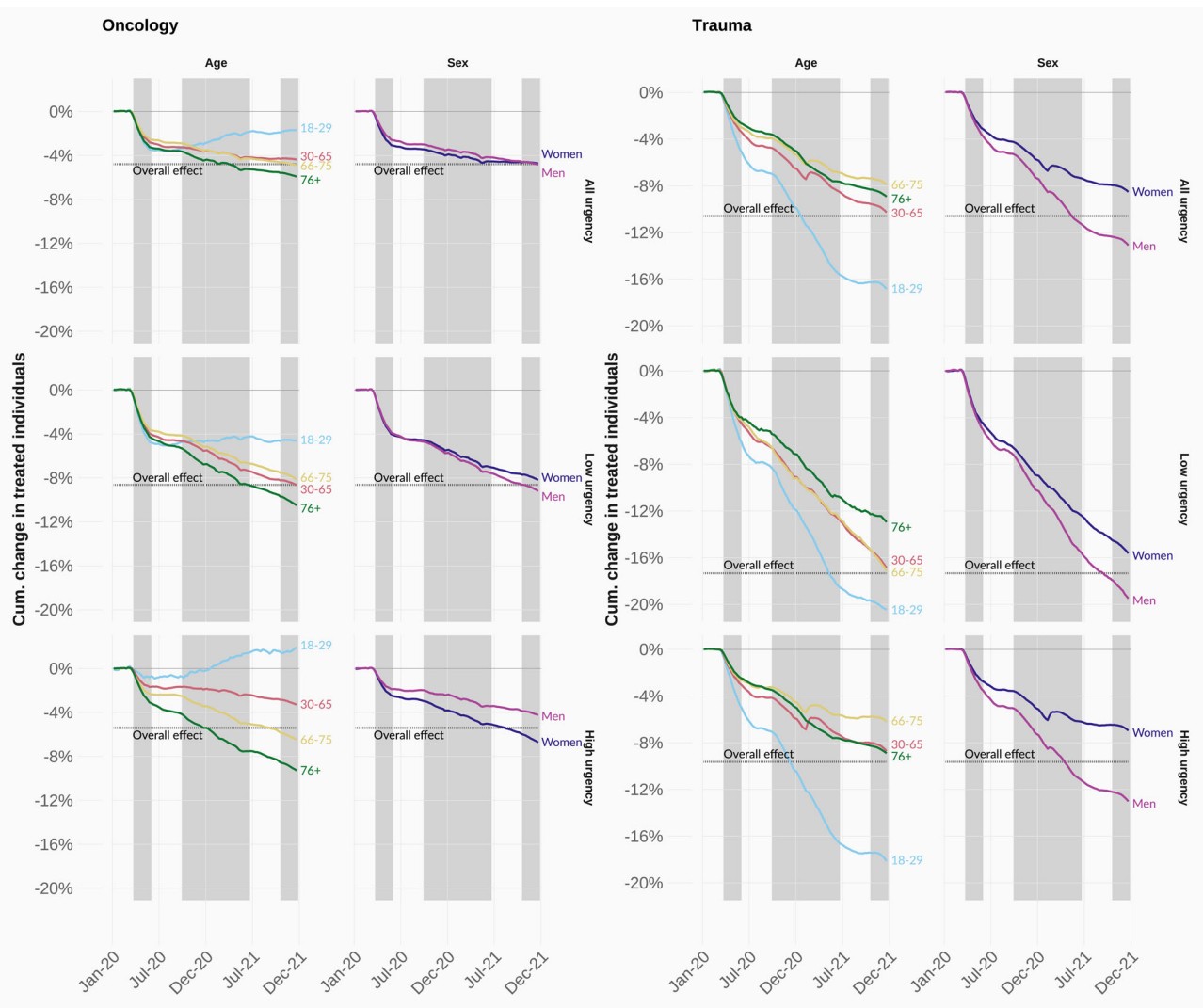

**Fig. 4 | Selected cumulative age- and sex-adjusted difference between the observed and predicted number of treated individuals in 2020 and 2021, across urgency types (rows) for a selection of sociodemographic groups (columns), for treatments related to oncology care (left) and trauma care (right).** COVID hospital waves are depicted in shaded grey. Full results can be found in Figures SI-3a and SI-3b.

## Discussion

The COVID-19 pandemic caused considerable strain on healthcare systems as the world battled a novel and ever-evolving disease. In anticipation of COVID hospitalisations and in order to limit further transmission, non-essential care was postponed. In parallel, government interventions were put in place to limit the spread of COVID. In The Netherlands, this meant partial lockdowns from March to May 2020, and again from October 2020 to June 2021 and towards the end of 2021. Together, these events resulted in declines to non-COVID healthcare procedures, a pattern that is observed worldwide. We contribute to this literature by analysing how reductions to non-COVID healthcare have been distributed across sociodemographic groups. Our study capitalises on unique full-population insurance data from the Dutch universal health coverage system and population register data to document the total decline in healthcare use and to examine how this decline differed by sociodemographic characteristics.

We estimate that in the early peak of the pandemic (mid-March to mid-May 2020) there were weeks with more than 300,000 people not

receiving care in the Netherlands, which amounts to close to 45% fewer individuals being treated. By the end of 2021, there were more than 4 million fewer patients treated than expected, a 6% reduction from prior years. The declines in healthcare procedures are not offset by the number of individuals entering the Dutch healthcare system for COVID, demonstrating the considerable strain of the pandemic on the standard provision of healthcare. We also find that although this decline is most pronounced for non-critical procedures, both less-urgent and highly-urgent procedures declined in 2020 and 2021: there were 9% fewer patients seen for low-urgency procedures, compared to 5% for highly urgent procedures. These numbers illustrate the challenges of adapting a healthcare system to a novel disease, which included increased care requirements (the average COVID-19 hospital stay was 8.2 days, compared to 5.9 days for a typical non-COVID procedure involving a hospital stay), policies to limit transmission to other patients (including partial closures of hospitals), as well as elevated sick leave among staff. Still, despite the fact that considerable adaptation occurred to some of the new challenges of COVID-19, losses continue throughout 2021, particularly for less urgent procedures. By the end of 2021, another one million fewer weekly patients had received care, in addition to the decline of more than three million weekly patients in 2020.

Importantly, we find strong differences between socio-demographic groups in the reduction in care. Urgent and less urgent procedures were more likely to be missed by those with a migrant background, people living below the poverty line, and older people. We also find striking differences across demographic groups when separately assessing medical procedures: declines in trauma were mostly concentrated among young males between the ages of 18 to 29. Reductions in highly urgent oncological care were almost twice as large for women compared to men and considerably larger for the older population, whilst remaining stable throughout 2020 for 18 to 29-year-olds and slightly increasing in 2021.

The policies put in place to limit hospitals from being over-whelmed due to COVID hospitalisations should not have affected sociodemographic groups differently. This makes the social inequal-ities that we observe particularly concerning. Although access to healthcare is universal, the Dutch system consists of various types of healthcare providers. Some require patient outreach to utilise, such as finding clinics with capacity for certain procedures and contacting those clinics to sign up for care. More privileged individuals may have experienced fewer barriers to access these types of providers than marginalised groups or have benefited from increased health literacy.

Our findings are especially troubling as the potential for inequality is likely smallest in countries with universal healthcare systems, like The Netherlands. Other healthcare systems, such as the United States, rely heavily on both private insurance and private providers, which intro-duce inequalities in both the access to and navigation of care. The Netherlands could be viewed as a best case scenario with excellent public infrastructure, near-universal broadband coverage, a healthy population, and an advanced and equitable healthcare system. By demonstrating that the pandemic disproportionately impacted health use among marginalised population groups in an otherwise equitable healthcare system, we contribute to a growing body of literature on the unequal nature of the COVID-19 pandemic that goes beyond its direct toll on mortality and disease burden[36–39]. We speculate that other nations with less equitable healthcare systems will have witnessed greater dis-parities in healthcare use during the pandemic, and we encourage fur-ther research to examine how our findings differ internationally.

Besides differences in access, we also find suggestive evidence for other behavioural mechanisms that may have led to the observed declines in healthcare usage, as well as differences among socio-demographic groups. Strong declines in highly urgent trauma-related care, especially among young men, likely reflect the effects of gov-ernment lockdown and subsequent declines in events that might lead to the need for trauma-related care. In the spring of 2020 and 2021, all public events and large gatherings were prohibited across The Neth-erlands and in the winter of 2020, a stringent lockdown was imple-mented, where most non-essential facilities were closed down. These periods coincide with reductions in highly urgent trauma-related care.

At the same time, we find troubling evidence for declines in healthcare procedures that should not have materialised, including a 5% decline in highly urgent oncological treatments. Although it is reassuring that among the young (18–29-year-olds) we observe almost no reduction in the number of weekly patients receiving highly urgent oncological care by the end of 2020, the observed differences between the old and the young and between females and males are worrying. While pausing some oncological treatments, such as screenings, might have been a necessary decision at the time, there is potential that it may induce excess deaths from cancers in the future[40]. Our findings of declines in non-urgent oncological care are consistent with findings in The Netherlands, as well as globally, of a pandemic-induced cancer screening backlog[22,41,42] 2021. We already see some possible signs of the downstream consequences of missed cancer screenings in the form of a considerably stronger decline in high-urgency oncological care among females compared to males, which could be a result of dis-ruptions to the nationwide screenings for breast and cervical cancer

during COVID-19 that are unique to females. In The Netherlands, screenings for breast cancer were stopped at the beginning of the pandemic and only recommended in an altered form in July 2020, relying more on self-testing rather than screenings performed by general practitioners[43]. Similarly, we find that among 18 to 29-year-olds there has been a relative increase of highly urgent oncological care in 2021 that might have been a result of missed screenings during 2020.

Another possible explanation for both the overall declines in non-COVID healthcare as well as the sociodemographic differences in these declines could be hesitance to seek care or differences in health lit-eracy. Individuals were weary of entering into the healthcare system during the pandemic for fear of infection[8,27–29], making it plausible that at least part of the decline in healthcare use can be explained by individuals not seeking the care they need. Our finding that young individuals were considerably less affected by reductions in care and were more likely to receive nonurgent healthcare compared to older individuals is consistent with differing risk assessments regarding COVID infections for different age groups.

The pandemic has affected healthcare through a myriad of pro-cesses, ranging from institutional practices that have limited the sup-ply of healthcare, government interventions that have altered the nature of healthcare demand, and more general behavioural changes on healthcare use. We exploit unique data encompassing all individual-level healthcare procedures in The Netherlands linked with rich sociodemographic variables to provide the first complete assessment of declines in healthcare use during the pandemic by socio-demographic characteristics. We find that sociodemographic groups differed considerably in their reduction of healthcare use. These findings are important for policymakers as they continue efforts to make up for delayed care, but also in better understanding the col-lateral impact of health crises like the COVID-19 pandemic beyond its direct toll on health and well-being.

Further work is necessary to better understand how these differ-ences in healthcare use have materialised. For example, our data does not include information on care provided by general practitioners (GPs), who are often the first point of care and typically provide gui-dance on whether to send patients to specialist care at a hospital. Studying care practices at the GP level could provide further insights into possible inequities in access, and the extent to which observed declines in healthcare procedures occur as a result of hesitancy from GPs – who may have opted to reduce their referrals to the hospital – or from hesitancy and a lack of health literacy from potential patients – who may have opted not to seek care. Similarly, we only measure use of the healthcare system and not objective need. Complementing observed use with need would provide further pointers to disentangle the extent to which avoidance of care has driven overall declines and differences among sociodemographic groups.

Another crucial avenue for further research is to evaluate the mid-and long-term health impact of declines to non-COVID healthcare. Already at the time of writing, the most recent analyses by the Dutch registry indicate that excess mortality at the end of 2021 was higher than COVID-19 mortality, especially among those between the ages of 65 and 80[44]. In 2022, excess mortality was again highest among older people[45]. This is broadly in line with our findings that older people suffered the strongest reductions in non-COVID healthcare use. However, for many disease burdens the full effects on adverse health outcomes will only materialise with time. It is paramount to understand how reductions in healthcare use may lead to adverse downstream health outcomes in the future, in order to be better prepared for future health crises. Our findings provide a starting point for such assessments.

Our study serves as a reminder that the health consequences of the COVID-19 pandemic span beyond mortality and long COVID, to include a profound impact on non-COVID-related healthcare use. Importantly, we show that the burden of declines in healthcare use has not been distributed equally, providing important lessons for future

health crises. Although our study takes place in the Dutch context, where the universal healthcare provision strives to minimise healthcare inequalities, we still observe strong inequities in reduced healthcare. These disparities are likely considerably smaller than in other contexts, where baseline access to healthcare is systemically unequal, such as in the United States. As policymakers and healthcare professionals strive to catch up on missed care, it is critical to understand that targeted efforts to reach historically marginalised and disadvantaged population groups are of utmost importance.

## Methods

Our research complies with all ethical regulations and was approved by the Research Ethics Committee of Department of Sociology at the University of Oxford (reference code: SOC_R2_001_C1A_21_64).

We use Dutch registry data, accessed through the Remote Access environment hosted by the Centraal Bureau voor de Statistiek[46]. The registry contains individual-level records of all persons residing in The Netherlands each year, including detailed information on their social, demographic, and economic background (see Table SI-2 for descriptive statistics of the population under study and Table SI-3 for a description of the sociodemographic variables we include). In addition, the registry contains information on every single health expenditure that was paid for by the universal health insurance system in The Netherlands. This includes all care that has been performed in hospital and was covered by the Dutch universal healthcare system. The dataset does not include data from general practitioners, as information on general practitioner care was not available, nor does it include data on non-primary care like dental treatment and physiotherapy.

Every activity covered by the universal health care system is logged and assigned various classifications, including the Diagnosis-Treatment-Classification (DTCs) that an activity falls under. The DTC reflects a substantive diagnosis and treatment plan and allows us to assign activities to medical groups like Oncology or Trauma. Each activity is time-stamped at the daily level and can be assigned an urgency level through the classification of DTC's by the Dutch Healthcare Authority (NZa). There are seven levels of urgency, ranging from extremely urgent (with a planning window of less than 24 hours) to non-urgent (with a planning window greater than 3 months). We assign procedures to one of three levels of urgency, with 'high urgency' procedures having a planning horizon of less than a week, 'middle urgency' procedures having a planning horizon of between one week and less than two months, and 'low' urgency procedures a planning horizon of two months or more. Some DTC's were not deemed necessary to classify by the NZa and are classified under 'no urgency'. Finally, DTC codes further classify activities into medical subgroups. We use these DTC codes to distinguish activities that fall under oncological care and trauma-related care. An example of highly urgent oncological care is "hospitalisations due to a malignancy in the brain" (NZa product code 29799070), which was the most frequently occurring treatment type in Oncology for 2019 besides general post-surgery care. An example of highly urgent trauma-related care is "surgery for an extra-articular trauma to an extremity" (NZa product code 199299069), which was one of the most frequently occurring treatment types in 2019, besides general trauma-related activities.

Yearly counts of the number of activities and treated individuals can be found in Tables SI4 to SI-7. Note that we prefer to study individuals treated in the system rather than the number of activities performed, as administrative changes occur in how activities are logged in the system from one year to the next. This is evident from the slight increase in the overall number of activities in 2018 and 2019 relative to 2017. An example of a year-on-year administrative change is the reclassification in 2021 of blood transplants to differentiate whether the donor was a family member or not, whereas no such distinction was made in the years prior. Although these yearly differences are easily modelled using year-fixed effects (see Fig. 1), we prefer to use the weekly number of treated individuals for consistency. We consider the weekly number of patients as the number of unique individuals having received at least one healthcare activity during that week. Note that we considered raw counts of activities as a robustness check (Figures SI-6a and SI-6b).

In our analytical approach, we calculate the number of unique individuals for each age and sex that had at least one activity performed in a given week for the period 2017-2021. This generates a set of weekly counts of unique individuals that were treated for every combination of age and sex. Note that if an individual seeks treatment in two weeks in a given year, she is counted twice. We calculate these timeseries for various subsets of activities and sociodemographics. We then age-sex standardise the number of individuals treated per week to the full population in 2018, to ensure that our findings are not driven by differences in age and sex composition between groups and over time.

To generate estimates of the expected number of individuals that would have been treated in the period between March and December of 2020, we estimate a linear model using the weekly counts of individuals as the outcome based on data up until and including February 2020. Our model includes week-fixed effects, and the number of holiday days. We also include year-fixed effects. We use this model to make predictions for every week between March and December 2021. Note that whilst we predict healthcare use for all weeks between March and December, we exclude week 53 from the plots since those weeks include an irregular number of days. We estimate separate models for every sociodemographic and/or activity subset, for example, when considering the number of individuals with a non-Western background that made use of low-urgency activities. This means that we make predictions per sociodemographic group, based on pre-pandemic trends and compare that group's observed number of treated individuals with the group-specific prediction. In our robustness checks, we estimate the same model but using a Negative Binomial regression.

Finally, we use our estimates of the weekly number of individuals receiving care in the system and compare these with the observed number of weekly individuals. We show the differences between the expected and realised number of treated individuals as a weekly difference, as well as a cumulative sum over successive weeks. Note that we exclude all healthcare activities that are associated with a COVID-19 infection. We identify these activities as those DTC's that were assigned a COVID-19 ICD-10 code. To aid interpretation, we classified three distinct waves of COVID-19 hospitalizations, which we identified as periods with a consistently high number of individuals receiving treatment for COVID-19 (more than 1,500 unique individuals weekly). This leads to three distinct periods: i) weeks 12 through 22 of 2020, ii) week 39 of 2020 through week 25 of 2021, and iii) week 42 and onwards of 2021.

To assess the multivariate impact of sociodemographic variables on declines in health care use, we generate 32 unique weekly timeseries: one for each fully interacted sociodemographic group (e.g., 18–29-year-old females living above the poverty line without a migrant background). We then proceed to model these timeseries for the period 2017–2021, including week, year and holiday fixed effects, sociodemographic covariates, random effects at the socio-demographic group level and a binary variable for whether the week fell during the pandemic or not. The pandemic binary variable reflects the average decline in the weekly number of treated patients. Note that we use the same age-sex standardised counts as throughout the rest of our results. By interacting sociodemographics with the various pandemic dummies, we can assess the joint effect of these characteristics on declines in healthcare relative to a reference group whilst controlling for other sociodemographics (e.g., comparing the timeseries of individuals living below the poverty line versus those that do not). Counts for each fully interacted group at the start of 2020 are provided in Table SI-8 and descriptives for the dataset used for multivariate regression can be found in Table SI-9.

## Reporting summary

Further information on research design is available in the Nature Portfolio Reporting Summary linked to this article.

## Data availability

All results presented here are calculated from non-public registry data from Centraal Bureau voor de Statistiek (CBS), accessed through the Remote Access environment. CBS was not involved in the calculation of any of the results presented here. While the data are not publicly available, academic institutions can apply for access to the Remote Access environment through the CBS. The underlying data cannot be shared outside of the Remote Access environment as it consists of individual-level, privacy sensitive data. To access the data, an institutional license is required for access to the registry and an additional application has to be submitted for access to the medical claims data with VEKTIS (for additional information, see https://www.cbs.nl/en-gb/our-services/customised-services-microdata/microdata-conducting-your-own-research). Source data are provided with this paper.

## Code availability

All code underlying our analyses are available at: https://github.com/MarkDVerhagen/Dutch_healthcare_inequalities_COVID19.

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

## Acknowledgements

We acknowledge funding from ZonMw (Grant 10430252210004 and Grant 10430372310025), the Leverhulme Trust (Grant RC-2018-003) for the Leverhulme Centre for Demographic Science (AF, AMT, MV), the European Research Council grant ERC-2021-CoG-101002587 (AMT), the UK Research and Innovation (UKRI) under the UK government's Horizon Europe funding guarantee EP/X027678/1 (AMT). Previous versions of this manuscript benefited from feedback provided by the Leverhulme Centre for Demographic Science's Health Inequality Working Group. The content of this manuscript is solely the responsibility of the authors and does not necessarily represent the official views of the ERC, ZonMw, the UKRI, the Clarendon Fund, or the Leverhulme Trust.

## Author contributions

A.F., A.M.T. and M.V. conceptualized the paper. A.F. and M.V. curated data and conducted formal analyses. A.F., A.M.T., and M.V. wrote the first draft and contributed to the visualization, writing, and editing, and all read and approved a final version of the manuscript.

## Competing interests
The authors declare no competing interests.
