## [Peer Review File · Nature Communications]

Inequalities in healthcare use during the COVID-19 pandemicREVIEWER COMMENTS

Reviewer #1 (Remarks to the Author):

Dear Authors,

Although I have no concerns at all with the design of this study, the research question and results are to me, I fear, frankly, not that interesting.

You state in the introduction that the pandemic put an "immense strain" on the health care system that reached far beyond the need for covid care (and what this strain consisted of is, by the way, not made clear, until perhaps in the discussion). And understanding why care for around 50 patients in your data was cut for every 1 covid patient (and the health consequences of that - which, NB, is an empirical question, not at all given one) may perhaps be more crucial than understanding why it was 53 for one group and 48 for another...?

And re the strengths of your associations: The reader should note that an e.g. 1,2 higher risk for 76+-year-olds compared to 18-29-year-olds of missing "a highly urgent treatment", in your study, means that a vast majority then, or 100 out of 120 affected 76-year-olds, would have been affected even if they had the same low risk as the 20-year-olds. To put the relative risk of 1,2 into perspective, at the same time, the relative risk of dying from covid for 76+-compared to 20-year-olds, was around, say, what, 200?

I am sorry to be a drag, but I am sure that you will be able to capitalize on your nice approach and data to study say different causal effects of health care, and perhaps empirically show which care should have been given anyway, and which was indeed rational to wait with.

Two more minor comments:

You list three reasons to expect group differences in declined health care use (needs, avoiding care (or the "hospital" as you, a bit too simplified, I guess, put it), and difficulties in navigating the system), but surely cuts in the supply will have mattered too: No health problem is distributed equally, so every planned cut back of care will also strike unequally, no?

Although The Netherlands may indeed present a "unique" context (densely populated and an efficient healthcare system), I fail to grasp why this should make it an ideal candidate for studying inequalities. What are we learning from The Netherlands that we could not be learning from another context? You dwell somewhat more on this in the discussion, but I miss it in the beginning.

I wish you all the best and good luck with further endeavours,

Anton Lager

Reviewer #2 (Remarks to the Author):

- My main comment is that the data is only up until December 2020, for many population groups uptake of healthcare has likely returned to pre-pandemic rates, more recent data would be more helpful to focus strategies for reducing inequalities on groups that continue to experience reduced access to healthcare.

- The introduction, while interesting and informative, could be shortened

- Figure 2 and 4: Is the baseline specific to individual groups or the population baseline (ie did all urgency oncological procedures reduce by ~7% for males compared to the estimated number of males receiving oncological treatments based on 2017-2020 data or the population average of all urgency oncological treatments?)

- "However, we also document stark differences in the use of oncological procedures between

sociodemographic groups, particularly for critical procedures. By the end of 2020, females were almost twice as likely to miss a critical oncological procedure compared to males, and elderly individuals (76 or above) were 14 times less likely to receive critical treatment compared to younger individuals (18–29)”. An interesting point for discussion/ further research - was the reduction in highly urgent oncological procedures (or even all urgency oncological procedures) reflected by delayed diagnosis/ poorer prognoses/ worse outcomes of oncology patients in these groups in the years following.

- In addition to discussing hesitancy to seek healthcare could include discussion point about health literacy, understanding what is symptoms justify presentation despite the risk of covid infection, is this an underlying factor explaining how declines in healthcare use unequally affected certain populations

Reviewer #3 (Remarks to the Author):

The authors addressed an important question around the indirect health impacts of COVID. The quantitative evidence presented suggests disparities in subpopulations exist even in a country where healthcare is considered highly accessible.

___Presentation of the manuscript___

- The first six paragraphs of the Introduction were very well written, covering various plausible hypotheses which goes on to be addressed in the analysis. However, paragraph 7 onwards appears to look more like Results and Discussion.

- The Results section needs more structure to help readers understand which points are being addressed in the different paragraphs. The authors may consider following the flow of hypotheses presented in the Introduction and the use of explicit headers to help with this. The authors should also ensure that results presented around the same observations are not scattered in multiple places to avoid confusing the readers.

- Figure SI9 appears to be the foundation of all claims in this work and therefore is an important figure. I suggest moving this to the main text, perhaps, as a panel of Figure 1.

- Figure 3 does not add much information beyond what is already present in Figure 2.

- The following text seems to refer to Figure 1. Would be good to reference the figure to help readers connect between them.

“At the height of hospital use from COVID (the week of March 23rd, 2020), there were more than 6,300 additional individuals that received treatment for COVID infections, compared to a simultaneous decline...”

___Comments on modelling approach___

Although signals in the data are strong and alternative modelling approaches are unlikely to change the conclusions of this study, the modelling approach used could be improved to provide more appropriate estimates of the effects under study.

- While modelling each demographic and/or activity subset separately would allow for trivial predictions of the time-series as done in Figure SI9, it would be good to also perform multivariate regression to adjust for effects of the different covariates. The model should also explicitly model the effects of the pandemic to obtain estimates and uncertainties of the risk ratios. These adjusted quantities would be invaluable for the community.

- According to the script “src/functions.R” in the provided GitHub repository, the authors used

function `lm` to fit the regressions which assumes normal distribution in the dependent variable which may not be the most appropriate for count variables. Have the authors considered negative binomial regressions with number of individuals present incorporated as offsets to normalise for varying demography across years as opposed to assuming demography of 2018 (the current approach)?

____Further comments____

- Aside from the clear and non-controversial unevenness of health systems in the United States, it would be helpful if the authors could discuss implications of their findings in other social contexts, for instance, places with strong gender inequality issues or places with higher percentages of the population being younger than the Netherlands.

Response to Reviews

Subject: Revision (NCOMMS-23-20056) entitled "Inequalities in Healthcare Use during the COVID-19 Pandemic"

Dear Reviewers,

Thank you for reviewing our manuscript entitled "Inequalities in Healthcare Use during the COVID-19 Pandemic". We appreciate the thorough and helpful review of our manuscript and are more than happy to address the expressed concerns.

The suggestions of the reviewers were extensive, but fair and resulted in a major revision of the article and inclusion of multiple new analyses included in the main text and appendix. We are confident that given our extensive and serious revisions, we were able to meet the concerns of the Editor and reviewers.

We provide our detailed response below. On the following pages we document our point-by-point response to the reviewers' comments, cross-referencing similar feedback accordingly. We indicate reviewer/editor feedback in text beginning "C," our responses in text beginning "R", and the manuscript changes in response in "quotes". We have indicated changes in the revised manuscript with red font. We believe that our manuscript is strengthened because of this review and hope that the Editor and the reviewers agree.

We also point all to our repository, where all scripts and graphs are available:

https://github.com/MarkDVerhagen/Dutch_healthcare_inequalities_COVID19.

We are of course open to any additional suggestions you might have and look forward to receiving your feedback.

Sincerely,
Authors

REVIEWER COMMENTS

Reviewer #1 (Remarks to the Author):

C1. Although I have no concerns at all with the design of this study, the research question and results are to me, I fear, frankly, not that interesting.

R1. We appreciate the reviewer's candid feedback. We do, however, wish to impress upon them the importance of identifying inequalities in healthcare. Inequalities in missed care can set in motion a chain reaction, leading to inequalities in health, and, ultimately, inequalities in mortality. By identifying the places where the Dutch healthcare system may have missed patients, we are pinpointing areas where improvements are needed and are contributing to active policy efforts at catching up on missed care (National Health Authority, 2021). In addition, understanding how crisis situations can exacerbate inequalities in healthcare is critical to improving policy responses in the future.

C2. You state in the introduction that the pandemic put an “immense strain” on the health care system that reached far beyond the need for covid care (and what this strain consisted of is, by the way, not made clear, until perhaps in the discussion). And understanding why care for around 50 patients in your data was cut for every 1 covid patient (and the health consequences of that - which, NB, is an empirical question, not at all given one) may perhaps be more crucial than understanding why it was 53 for one group and 48 for another...?

R2. Thank you for raising these concerns. We sincerely hope that we did not paint a picture that problematizes focusing on COVID-related care, as this was most definitely not our intention. Instead, what we aim to do is highlight that there are inequalities in the disruption of non-COVID care. This, to us, is not placing any value judgment on the difficult decisions that healthcare providers made to focus on COVID-related care.

We agree that the extent to which non-COVID care was reduced with respect to the amount of COVID care that occurred in the system is striking. That said, others have already identified a considerable “substitution effect” during the first COVID wave. We feel the novelty of our data and results is the specific angle of inequality, and how these considerable declines in

non-COVID care were unequally distributed, as this is not broadly known and difficult to empirically assess without the kind of rich, administrative data that we use.

We do recognize that the reviewer is interested in the substantial size and nature of this substitution effect, and we discuss it in more detail in our revised manuscript (Discussion section, paragraph 2). In particular by adding 2021 data to our analyses, we are better able to highlight the considerable reduction in this substitution effect in the second and third COVID waves, suggesting that the healthcare system adapted to the additional strain. However, we feel it is beyond the scope of this paper and perhaps leading away from our key contribution to dive into why this substitution effect was so high initially. We do agree with the Reviewer that this is an important empirical fact that deserves repeating, and that the changes in this substitution effect are also noteworthy and should prompt general introspection. Therefore, we would like to thank the Reviewer for pushing us to discuss this feature of the data in more detail in the revised manuscript.

The reviewer's concern about the language of "immense strain" in the introduction is well-received. We have moved the paragraphs that include this language exclusively to the discussion. This is also in line with comments from Reviewer 2 (C8) and Reviewer 3 (C13) that our introduction is too long. We hope the reviewer finds the placement of this section better suited for the discussion.

C3. And re the strengths of your associations: The reader should note that an e.g. 1,2 higher risk for 76+-year-olds compared to 18-29-year-olds of missing "a highly urgent treatment", in your study, means that a vast majority then, or 100 out of 120 affected 76-year-olds, would have been affected even if they had the same low risk as the 20-year-olds. To put the relative risk of 1,2 into perspective, at the same time, the relative risk of dying from covid for 76+-compared to 20-year-olds, was around, say, what, 200?

R3. The reviewer is correct that we could have better explained how to interpret relative risk values. In fact, we think relative risk might be unnecessarily complicated and, as also mentioned by Reviewer 3 (C16) there is not much added in the relative risk metrics originally presented in Figure 3 as compared to the information that is available in Figure 2 already. As such, we have removed the relative risk metrics from the revised manuscript. Instead, to reduce possible confusion, we describe proportional declines within groups only, e.g.:

“By the end of 2021, individuals below the poverty line had experienced reductions in low and medium urgency procedures of 10.4% and 6.1% respectively.” (Results section, sub section *Sociodemographic inequalities in non-COVID healthcare use declines*, paragraph 1)

C4. I am sorry to be a drag, but I am sure that you will be able to capitalize on your nice approach and data to study say different causal effects of health care, and perhaps empirically show which care should have been given anyway, and which was indeed rational to wait with.

R4. We understand the desire for causal estimates and rational choice analyses, but this is beyond the scope of the present study: to identify whether inequalities occurred in the declines in non-COVID healthcare use during the pandemic. Moreover, the true causal effects of (at least some of the) missed healthcare will likely lie further in the future (e.g., the long-term consequences of reduced cancer screening), for which adverse outcomes are not (yet) available.

However, we share the reviewer's interest in understanding whether the effect of the pandemic on treatment use differed regarding the necessity of those treatments. We address this by assigning health activities to urgency levels, and our results indeed show that healthcare use declines across both very urgent and less urgent procedures but considerably more so among the latter. We thank the reviewer for pushing us to clarify this perspective, and we now highlight this more explicitly in the Discussion section (paragraph 10) of our revised manuscript where we discuss the long-term effects of declines in healthcare in the context of future work.

C5. Two more minor comments:

You list three reasons to expect group differences in declined health care use (needs, avoiding care (or the “hospital” as you, a bit too simplified, I guess, put it), and difficulties in navigating the system), but surely cuts in the supply will have mattered too: No health problem is distributed equally, so every planned cut back of care will also strike unequally, no?

R5. This is a good point, and we thank the reviewer for raising it. However, we remind the reviewer that our analytic approach projects forward the anticipated healthcare use from prior years and we express our main results in proportion to this expected need. That said, the reviewer is correct in that hypothetical cuts to activities that are almost entirely utilized by certain demographics could lead to inequality. In fact, we observe possible evidence of this in reduced breast and cervical cancer screenings that will have affected only females in our study population. These types of effects are what we consider to be part of the hospital (note below that we change this language to “healthcare facilities”) and can be understood as decision-making among healthcare providers in how to reduce care.

We agree with the reviewer that this perspective should have been fleshed out more, which we now do in the paragraph headed “Declines for trauma-related and oncological care” in the Results section, as well as amending the rather simplistic terminology in paragraph 4 of our introduction (as mentioned by the Reviewer), which now reads:

“Second, people may have avoided visiting **healthcare facilities** during the pandemic, resulting in missed care.” (Introduction section, paragraph 4)

We also make a similar point regarding pre-pandemic differences in healthcare use as a driver of inequalities:

“Such injuries are more common for some population groups than others, so changes to health needs may have contributed to differences in healthcare use between population groups.” (Introduction section, paragraph 3)

C6. Although The Netherlands may indeed present a “unique” context (densely populated and an efficient healthcare system), I fail to grasp why this should make it an ideal candidate for studying inequalities. What are we learning from The Netherlands that we could not be learning from another context? You dwell somewhat more on this in the discussion, but I miss it in the beginning.

R6. Thanks for drawing our attention to our poor justification for The Netherlands in the introduction/background. We now elaborate on this in paragraph 7 of the revised manuscript.

“The Netherlands presents a unique context for analysing inequalities in healthcare use during the pandemic, as healthcare coverage is universal and publicly available to all residents. The Netherlands is also characterised by i) small land mass, high population density, and dense infrastructure, ii) an efficient healthcare system and healthy population base, and iii) a wealthy population with comparatively low income inequality. This makes The Netherlands a case study where there is little reason to expect substantial inequalities in access to healthcare services.”

To provide further context to our logic for why The Netherlands is unique and important for studying inequalities in healthcare declines, we offer some additional justification here. Universal healthcare systems are designed such that (in theory) all people should receive equal access to care. There are many Western European nations that have universal healthcare systems. What differentiates The Netherlands from these other nations is its densely-populated small landmass, making the distance to healthcare facilities much shorter than other nations.

There is extensive literature on how the distance to healthcare facilities can affect the likelihood of using them (see Kelly et al. 2016 for a systematic review).

Thus, as we mention in the introduction (Page 3, Paragraph 3), “The Netherlands serves as a best case scenario for a resilient and equitable healthcare system” [text from original draft]. Any observation of inequalities in healthcare during the pandemic would indicate factors beyond the cost of using healthcare might be at play. Akin to our response above (R1), we note that these inequalities are important to identify because it can give the Dutch government knowledge of where to target its efforts at reducing persistent health inequalities. We thank the reviewer for being more explicit about the relevance of our case study earlier in our manuscript.

Reviewer #2 (Remarks to the Author):

C7. My main comment is that the data is only up until December 2020, for many population groups uptake of healthcare has likely returned to pre-pandemic rates, more recent data would be more helpful to focus strategies for reducing inequalities on groups that continue to experience reduced access to healthcare.

R7. We agree wholeheartedly with the reviewer here, and appreciate their patience as we waited for a data update from The Netherlands. All analyses have now been expanded through December 2021, the most recent update available.

This new data allows us to identify that, as the reviewer suggests, declines in healthcare use have reduced considerably in the second COVID year. This further strengthens our assessment that the healthcare system learned how to manage the stressors of the pandemic. Furthermore, we find that inequalities persist beyond the first year and slightly widened as recurring COVID waves meant healthcare use did not return to pre-pandemic levels across 2021. We would like to thank the Reviewer for pushing this point, as we feel it greatly improves the relevance of the paper and the strength of our analyses.

We have updated our results, discussion, as well as abstract, to reflect these new data and analyses.

C8. The introduction, while interesting and informative, could be shortened

R8. Thank you for your candidness. Reviewer 3 shared a similar concern (C13). We have now amended our introduction in two ways. First, we edited the introduction to be more concise, as there were some places where we were a bit repetitive. Second, at the suggestion of Reviewer 3 (C13), we have moved the final three paragraphs of the introduction to the discussion. This, we hope, makes our introduction both shorter yet still informative.

C9. Figure 2 and 4: Is the baseline specific to individual groups or the population baseline (ie did all urgency oncological procedures reduce by ~7% for males compared to the estimated number of males receiving oncological treatments based on 2017-2020 data or the population average of all urgency oncological treatments?)

R9. Baseline specifications are specific to individual groups. So, in 2020, oncological procedures for males were ~7% lower than the estimated number of males receiving

oncological treatments based on data from January 2017 through Feb 2020. We have now made this point more explicit in the Methods section of our paper:

“This means that we make predictions per demographic group, based on pre-pandemic trends and compare that group’s realized number of treated individuals with the group-specific prediction.” (Methods section, paragraph 4)

C10. “However, we also document stark differences in the use of oncological procedures between sociodemographic groups, particularly for critical procedures. By the end of 2020, females were almost twice as likely to miss a critical oncological procedure compared to males, and elderly individuals (76 or above) were 14 times less likely to receive critical treatment compared to younger individuals (18–29)” An interesting point for discussion/ further research. Was the reduction in highly urgent oncological procedures (or even all urgency oncological procedures) reflected by delayed diagnosis/ poorer prognoses/ worse outcomes of oncology patients in these groups in the years following.

R10. We agree with the Reviewer that it is interesting to see how declines in particularly preventative diagnostic activities have resulted in adverse outcomes and this is a crucial avenue for future research. Unfortunately, the expected timeline for such adverse outcomes lie further in the future than our data currently permits us to assess.

We do think it is important to more actively discuss long-term effects of missed healthcare and include a discussion of excess mortality studies from the Dutch population registry that are considerably less granular, but have a shorter time lag and can shed some light on adverse health outcomes in the future. More generally, we discuss the long-term health effects of missed care at length in our revised Discussion section (Discussion section, paragraph 11).

C11. In addition to discussing hesitancy to seek healthcare could include discussion point about health literacy, understanding what is symptoms justify presentation despite the risk of covid infection, is this an underlying factor explaining how declines in healthcare use unequally affected certain populations

R11. This is an excellent point, especially since there is stratification in who has access to the resources. We direct the reviewer to our introduction, where we included this point in our original manuscript (Introduction section, Paragraph 5):

“It is already known that marginalised people frequently experience barriers to accessing the same quality healthcare as their more advantaged peers, a pattern that can manifest through

structural barriers (e.g., distance to healthcare facilities), financial barriers (e.g., cost of healthcare), or health beliefs and literacy (Levesque et al. 2013; World Health Organization 2010).”

We could have better incorporated this into our discussion though, and have added clauses about this in two places where we discuss hesitancy:

“Another possible explanation for both the overall declines in non-COVID healthcare as well as the demographic differences in these declines could be hesitancy to seek care **or differences in health literacy.**” (Discussion, Paragraph 7)

And

“Studying care practices at the GP level could provide further insights into possible inequities in access, and the extent to which observed declines in healthcare procedures occur as a result of hesitancy from GPs – who may have opted to reduce their referrals to the hospital – or from hesitancy **and lacking health literacy** from potential patients – who may have opted not to seek care.” (Discussion, Paragraph 9)

Reviewer #3 (Remarks to the Author):

C12. The authors addressed an important question around the indirect health impacts of COVID. The quantitative evidence presented suggests disparities in subpopulations exist even in a country where healthcare is considered highly accessible.

R12. We thank the reviewer for acknowledging the importance of our work, the relevance of our case study and for succinctly describing our major takeaways.

C13. The first six paragraphs of the Introduction were very well written, covering various plausible hypotheses which goes on to be addressed in the analysis. However, paragraph 7 onwards appears to look more like Results and Discussion.

R13. Thank you for noting this. Reviewer 2 (C8) also noted that our introduction was a bit too long. In line with the feedback here and that in C8, we have amended our introduction to be more concise and move the final three paragraphs of the introduction to the discussion.

C14. The Results section needs more structure to help readers understand which points are being addressed in the different paragraphs. The authors may consider following the flow of hypotheses presented in the Introduction and the use of explicit headers to help with this. The authors should also ensure that results presented around the same observations are not scattered in multiple places to avoid confusing the readers.

R14. We thank the reviewer for these suggestions. In line with the feedback, we have now edited the results section to (1) reflect the inclusion of new data and analyses (see C7), and (C18) to flow more succinctly, including the proposed adjustments to our graphs (see C15 and C16 below). We also removed some of the more repetitive results, like the original supplementary figures SI-4a and SI-4b which contained results for a subset of medical activities that were almost identical to the subsets presented in SI-2a and SI-2b. We also structured our the Results section into separate subsections.

We hope the reviewer now finds our Results section easier to follow.

C15. Figure SI9 appears to be the foundation of all claims in this work and therefore is an important figure. I suggest moving this to the main text, perhaps, as a panel of Figure 1.

R15. Thank you for for this suggestion. We have moved this to the main manuscript.

C16. Figure 3 does not add much information beyond what is already present in Figure 2.

R16. We agree with the reviewer and have removed Figure 3.

C17. The following text seems to refer to Figure 1. Would be good to reference the figure to help readers connect between them. “At the height of hospital use from COVID (the week of March 23rd, 2020), there were more than 6,300 additional individuals that received treatment for COVID infections, compared to a simultaneous decline...”

R17. Thank you for catching that. In our revision, we added a new Figure 1, making the figure the reviewer references here Figure 2. We have edited the text to refer to this accordingly. It now reads:

“At the height of hospital use from COVID (the week of March 23rd, 2020, see Figure 2), there were more than 6,300 additional individuals that received treatment for COVID infections, compared to a simultaneous decline in non-COVID related healthcare of more than 300,000 individuals.”

C18. Although signals in the data are strong and alternative modelling approaches are unlikely to change the conclusions of this study, the modelling approach used could be improved to provide more appropriate estimates of the effects under study.

A. While modelling each demographic and/or activity subset separately would allow for trivial predictions of the time-series as done in Figure SI9, it would be good to also perform multivariate regression to adjust for effects of the different covariates. The model should also explicitly model the effects of the pandemic to obtain estimates and uncertainties of the risk ratios. These adjusted quantities would be invaluable for the community.

B. According to the script “src/functions.R” in the provided GitHub repository, the authors used function `lm` to fit the regressions which assumes normal distribution in the dependent variable which may not be the most appropriate for count variables. Have the authors considered negative binomial regressions with number of

individuals present incorporated as offsets to normalise for varying demography across years as opposed to assuming demography of 2018 (the current approach)?

R18. We would like to thank the Reviewer for pushing us to further refine our analysis. We interpret Point A as a concern that there may be some confounding covariates within our result. For example, inequalities for those living in poverty may be confounded with other covariates like those with a Non-native migrant background. Although our paper is meant to be descriptive in nature, we agree that further untangling what is driving results in each subgroup is important.

As such, we have included an additional analysis where we generate time series for complete interactions of the covariate space and urgency classifications (e.g., low urgency activities for Poor, Dutch Native, Male, 30-65 year olds). This leads to a large number of distinct time series spanning the period January 2017 until December 2021. We then proceed to model these timeseries whilst adding the demographic characteristics of each time series as covariates. We also include, as suggested by the Reviewer, both a singular treatment effect as well as models including the different phases of the pandemic separately.

The results of this new analysis can be found in Figure SI-1. As the Reviewer expected, this alternative coding strategy does not meaningfully alter the results but does provide interesting additional insights, which we now highlight in our Results section:

“These findings are further supported by multivariate regressions where we assess the joint impact of demographic characteristics on weekly declines in non-COVID healthcare, as well as during different stages of the pandemic (see Figure SI-1). We find that associations between those below the poverty line and non-native Dutch people operate independent from one another. Further interactions between our sociodemographic variables and pandemic time periods show that females had the strongest decreases during the first wave. Relative to their male counterparts, they endured a 20 percentage point additional decline in weekly treated patients relative to pre-pandemic averages. From the second wave onwards, this pattern is reversed. Overall, the strongest inequality across the entire period up until December 2021 were experienced by non-native Dutch people.” (Results section, subsection Sociodemographic inequalities in non-COVID healthcare use declines, paragraph 2)

We would again like to thank the Reviewer for pushing us on expanding our analytical framework as we feel that we have obtained additional insights and are more secure in our original findings through it.

With respect to Point B, we include a robustness check of our main results when using a negative binomial regression regression to make weekly predictions. These can be found in Figures SI-9a and SI-9b:

“We also assess whether our results are robust to using a Negative Binomial regression to make estimates of expected healthcare use instead of a standard linear regression and find similar results (SI-9a and SI-9b).” (Results section, subsection *Robustness checks*, paragraph 2)

C19. Aside from the clear and non-controversial unevenness of health systems in the United States, it would be helpful if the authors could discuss implications of their findings in other social contexts, for instance, places with strong gender inequality issues or places with higher percentages of the population being younger than the Netherlands.

R19. Great suggestion. We now elaborate further in our discussion about what our findings might signal for other contexts including:

“We speculate that other nations, including those with gender equality concerns, will witness greater disparities in healthcare use during the pandemic, and we encourage further research to examine how our findings differ internationally” (Discussion section, paragraph 5)

Response References

Kelly, C., Hulme, C., Farragher, T., & Clarke, G. (2016). Are differences in travel time or distance to healthcare for adults in global north countries associated with an impact on health outcomes? A systematic review. *BMJ open*, 6(11), e013059.

National Health Authority. (2021). Kader 'passende inhaalzorg MSZ'.
<https://www.rijksoverheid.nl/documenten/publicaties/2021/05/26/kader-passende-inhaalzorg-msz>

REVIEWERS' COMMENTS

Reviewer #1 (Remarks to the Author):

Dear authors,

Thank you for lengthy additions, revisions and replies. Yet, all-in-all, my critique stands. The demonstrated overall differences between sociodemographic groups are hardly "strong" or "striking", as you claim in the discussion, but very small; and, in contrast to your reply of my first previous comment, unlikely to lead to any detectable future differences in morbidity or mortality. It does not help that you no longer actually present the low relative risks in the paper.

You state that "individuals living below the poverty line had experienced reductions in low and medium urgency procedures of 10.4% and 6.1% respectively, compared to 9.0% and 4.3% for those above the poverty line. We document a similar trend for individuals with a migrant background, who experienced reductions of 10.0% and 4.7% in low and medium urgency procedures compared to 9.0% and 4.3% for native Dutch individuals".

These differences translate into relative risks of 1.16; 1.42; 1.11; and 1.09 (which for example can be compared with the GRADE Handbook's definitions of "large" and "very large" effects, i.e. >2.0 and >5.0 , respectively (<https://gdt.gradepro.org/app/handbook/handbook.html>)) or Cohen's d s of 0.04; 0.08; 0.02; and 0.02 (which would translate into far from meaningful differences (cf Cohen 1988 or e.g. https://en.wikipedia.org/wiki/Cohen's_d)).

From "Figure 3: all urgency", it seems like sex differences have disappeared completely - as a result of the new longer follow-up. Over-and-above differences in certain subgroup analyses, this leaves a bit clearer differences by age, which I guess were wanted/in line with policies.

Best

Reviewer #2 (Remarks to the Author):

I am happy with the revisions and well done to the authors

Reviewer #3 (Remarks to the Author):

I thank the Authors for thoroughly addressing my comments. I have no further scientific concerns.

REVIEWERS' COMMENTS

Reviewer #1 (Remarks to the Author):

Dear authors,

Thank you for lengthy additions, revisions and replies. Yet, all-in-all, my critique stands. The demonstrated overall differences between sociodemographic groups are hardly “strong” or “striking”, as you claim in the discussion, but very small; and, in contrast to your reply of my first previous comment, unlikely to lead to any detectable future differences in morbidity or mortality. It does not help that you no longer actually present the low relative risks in the paper.

You state that “individuals living below the poverty line had experienced reductions in low and medium urgency procedures of 10.4% and 6.1% respectively, compared to 9.0% and 4.3% for those above the poverty line. We document a similar trend for individuals with a migrant background, who experienced reductions of 10.0% and 4.7% in low and medium urgency procedures compared to 9.0% and 4.3% for native Dutch individuals”.

These differences translate into relative risks of 1.16; 1.42; 1.11; and 1.09 (which for example can be compared with the GRADE Handbook’s definitions of “large” and “very large” effects, i.e. >2.0 and >5.0, respectively (<https://gdt.gradepro.org/app/handbook/handbook.html>)) or Cohen’s hs of 0.04; 0.08; 0.02; and 0.02 (which would translate into far from meaningful differences (cf Cohen 1988 or e.g. https://en.wikipedia.org/wiki/Cohen's_h)).

From “Figure 3: all urgency”, it seems like sex differences have disappeared completely - as a result of the new longer follow-up. Over-and-above differences in certain subgroup analyses, this leaves sa bit clearer differences by age, which I guess were wanted/in line with policies.

Best

Reply:

We thank the Reviewer for their views on our revised manuscript. It is unfortunate the Reviewer does not agree with us that our findings are meaningful, but we accept this is a matter of subjective interpretation of our results, rather than a critique of our methods.

That said, we understand that our findings might not be significant within the context of relative risks as typically understood by medical methodologists. Such relative risks are typically used when comparing outcomes across different groups or when assessing the evidence of experimental treatments or medicine. Conversely, we discuss what can be seen as difference-in-differences at population-level scale, and within a context where no medical intervention was performed. We do however agree with the Reviewer that re-alignment of difference by sex are noteworthy, as we note in our manuscript (Section *Sociodemographic inequalities in non-COVID healthcare use declines*, paragraph 1).

Reviewer #2 (Remarks to the Author):

I am happy with the revisions and well done to the authors

Reply:

We would like to again thank Reviewer 2 for their important initial feedback, which improved the paper considerably.

Reviewer #3 (Remarks to the Author):

I thank the Authors for thoroughly addressing my comments. I have no further scientific concerns.

Reply:

We would like to again thank Reviewer 3 for their important initial feedback, which improved the paper considerably.